# Rationally Designed Novel Antimicrobial Peptides Targeting Chitin Synthase for Combating Soybean Phytophthora Blight

**DOI:** 10.3390/ijms25063512

**Published:** 2024-03-20

**Authors:** Yue Ran, Kiran Shehzadi, Jian-Hua Liang, Ming-Jia Yu

**Affiliations:** School of Chemistry and Chemical Engineering, Beijing Institute of Technology, Beijing 100081, China; 3120221227@bit.edu.cn (Y.R.); kiranshehzadi@bit.edu.cn (K.S.)

**Keywords:** *Phytophthora sojae*, soybean, biopesticide, molecular dynamic simulation

## Abstract

Soybean phytophthora blight is a severe menace to global agriculture, causing annual losses surpassing USD 1 billion. Present crop loss mitigation strategies primarily rely on chemical pesticides and disease-resistant breeding, frequently surpassed by the pathogens’ quick adaptive evolution. In this urgent scenario, our research delves into innovative antimicrobial peptides characterized by low drug resistance and environmental friendliness. Inhibiting chitin synthase gene activity in *Phytophthora sojae* impairs vital functions such as growth and sporulation, presenting an effective method to reduce its pathogenic impact. In our study, we screened 16 previously tested peptides to evaluate their antimicrobial effects against *Phytophthora* using structure-guided drug design, which involves molecular docking, saturation mutagenesis, molecular dynamics, and toxicity prediction. The in silico analysis identified AMP_04 with potential inhibitory activity against *Phytophthora sojae*’s chitin synthase. Through three rounds of saturation mutagenesis, we pin-pointed the most effective triple mutant, TP (D10K, G11I, S14L). Molecular dynamic simulations revealed TP’s stability in the chitin synthase-TP complex and its transmembrane mechanism, employing an all-atom force field. Our findings demonstrate the efficacy of TP in occupying the substrate-binding pocket and translocation catalytic channel. Effective inhibition of the chitin synthase enzyme can be achieved. Specifically, the triple mutant demonstrates enhanced antimicrobial potency and decreased toxicity relative to the wild-type AMP_04, utilizing a mechanism akin to the barrel-stave model during membrane translocation. Collectively, our study provides a new strategy that could be used as a potent antimicrobial agent in combatting soybean blight, contributing to sustainable agricultural practices.

## 1. Introduction

Soybeans are one of the world’s most important crops, covering 6% of global arable land and supplying 60% of oilseed production [1,2]. Soybean seeds are rich in protein, oil, micronutrients, flavonoids, and saponins, have become vital crops for human and livestock consumption, and have extensive industrial applications [3]. Soybean plays a crucial role in meeting over 25% of the global demand for food and animal feed protein. It serves as one of the primary commercial crops for plant oils and protein, making it essential for ensuring global food security. According to the latest report from the International Grains Council, global soybean production has escalated from 73,481,964 metric tons in 1980 to 358,337,972 metric tons by 2023, yielding significant profits [4].

Nowadays, the annual damage caused by *P. sojae*, a pathogen responsible for stem and root rot, results in over USD 1 billion of losses in global soybean production [5]. *P. sojae* is a destructive soil-borne oomycete pathogen that is not easy to control, resulting in reduced soybean yield. Since its initial report in 1958, *P. sojae* has been identified as one of soybean’s most damaging and widely spread pathogens, affecting yield and quality from seedling to seed development stages [6]. It can lead to a 50% reduction in production or even no harvest. Consequently, researchers have looked into *P. sojae* prevention and control extensively. Currently, the control methods for *P. sojae* primarily involve genetically modified crops and pesticides. However, both of these control measures can pose risks to ecological health and the environment, such as environmental degradation, global warming, and the induction of carcinogenesis. Genetically modified soybeans may bring about various detrimental effects. Firstly, they have the potential to alter biodiversity, leading to the emergence of insects and weeds that are resistant to pesticides [7]. Secondly, excessive cultivation of genetically modified soybeans can trigger a series of unfavorable environmental problems, such as increased deforestation, global warming, and acidification of freshwater and land [8,9].

Moreover, several studies indicate that genetically modified soybeans may lead to reproductive toxicity in certain contexts or animal models [10,11]. At the same time, the excessive use of pesticides poses potential threats to ecosystems and human health. Pesticides exhibit persistence, bioaccumulation, and high environmental toxicity, persisting and accumulating in the food chain and potentially impacting organisms. Different pesticides exhibit varying toxicity to other species in the ecological environment. For human health, pesticide residues in food have been associated with increased risks of diabetes, lung cancer, and metabolic syndrome. Furthermore, pregnant women exposed to pesticides may transmit accumulated toxins to their babies through breast milk [12]. According to World Health Organization statistics, there are as many as 3 million cases of pesticide poisoning each year globally, making pesticide poisoning a global public health issue [13,14]. Antimicrobial peptides have several advantages compared to genetically modified soybeans and pesticides. One important point is that the multi-target interaction and mechanism of action of AMPs with the bacterial membrane makes the appearance of resistance to AMPs more difficult compared to conventional antibiotics [15]. AMPs predominantly target the bacterial cell membrane through electrostatic interactions, leading to membrane disruption via destabilization, changes in fluidity, and depolarization. This action is facilitated by the amphipathic nature of AMPs, which possess a balance of cationic and non-polar residues [16]. Unlike traditional antibiotics, which often target specific cellular processes and risk inducing resistance, the broad and nonspecific mode of action of AMPs makes it difficult for pathogens to develop resistance [17]. Additionally, some AMPs can penetrate cell membranes to target intracellular processes, further diversifying their mechanisms and reducing the likelihood of resistance [18]. Additionally, antimicrobial peptides exhibit broad-spectrum activity against various bacteria and have lower toxicity to host cells [19,20]. Being natural substances, they are considered safer and more reliable. Thus, the development of a green, environmentally friendly, and harmless antimicrobial peptide against *P. sojae* is crucial.

Notably, only a few drugs and one cyclic antifungal peptide from marine fungi have shown inhibitory effects against soybean phytophthora blight [21,22,23,24]. Interestingly, Fluoxapiprolin, Oxzthiapiprolin, and Ethaboxam, developed by Bayer and DuPont (Wilmington, DE, USA), have demonstrated the ability to inhibit the growth of *P. sojae* by targeting spores and mycelium (Figure 1) [25,26,27,28]. However, it is worth noting that these compounds belong to the thiazole class, and mutations in the single dominant genes (Rps) and *β-tubulin* have been identified as potential causes of drug resistance [29]. Moreover, the resistance of *P. sojae* to active pharmaceuticals has reduced the controllability of soybean root rot, which presents a severe risk to the cultivation of soybeans. Additionally, there is a lack of knowledge regarding the transmembrane mechanism of antimicrobial peptides and the mechanism underlying the interaction between antimicrobial drugs and target proteins. Therefore, there is an urgent need to develop antimicrobial peptides targeting soybean phytophthora to overcome drug resistance and unravel the mechanisms of their antimicrobial and transmembrane activities.

Antimicrobial peptides (AMPs) containing cationic and hydrophobic amino acid residues can cross cellular membranes by leveraging the interactive properties of these residues. They exert bactericidal effects by disrupting cell membranes or inhibiting intracellular biological processes. Upon pathogen attack, plant AMPs can be induced or expressed constitutively. AMPs exert their broad antimicrobial activity primarily by targeting structures such as pathogenic cell membranes, cell walls, biofilm formation, and hyphal formation, as well as influencing pathogenic factors [30]. Pathogenic cells differ from human cells, enabling selective targeting [31,32,33]. For example, plectasin inhibits the lipid II precursor in bacterial walls, demonstrating activity against *Staphylococcus aureus* without affecting human cells [34]. Chitin biosynthesis is necessary for the survival and propagation of many organisms from various taxonomic groups, including pest insects, oomycetes that ruin agriculture, and potentially fatal fungi. Therefore, inhibition of chitin biosynthesis is a desirable target for the discovery of pesticides or antimicrobial agents. Chitin synthase is a key enzyme that catalyzes chitin biosynthesis [35]. Research has shown that the inhibition or silencing of the chitin synthase gene substantially affects the growth, sporulation, and spore release of *P. sojae*, thereby reducing its pathogenicity. Since chitin is a vital component of *P. sojae*, targeting chitin synthase represents a potential breakthrough in controlling *P. sojae*. To address the control of *P. sojae*, we designed antimicrobial peptides targeting chitin synthase. Initially, we selected four peptides from a comprehensive antimicrobial peptide library, which contains 16 peptides that have been experimentally proven to exhibit inhibitory activity against various pathogens, including *Phytophthora*. The selected peptides exhibit various mechanisms to inhibit *Phytophthora*, including suppressing the output and germination of zoospores and zoosporangia [36], rendering zoospores immobile and causing them to lyse [37], affecting the expression of plant defensins [38], and disrupting and permeabilizing the inner membrane leading to cell death [39].

In this study, we identified the lead molecule AMP_04 for its high affinity to chitin synthase through virtual screening, and compared its binding mechanisms to the drug NikZ using molecular docking. AMP_04 nearly fills the substrate-binding site and catalytic channel of chitin synthase. Saturation mutagenesis led to a variant, TP (D10K, G11I, S14L), with enhanced affinity. Dynamic simulations showed TP disrupts cell membranes via an immediate rupture, akin to the barrel-stave mechanism, where its hydrophobic amino acids facilitate penetration, demonstrating its antimicrobial action [40]. 

## 2. Results

### 2.1. Virtual Screening Study

Initially, a screening was performed to select 16 antimicrobial peptides with known peptide sequences, which have been experimentally proven to exhibit inhibitory effects against *Phytophthora* species (Appendix A). We have selected the top four reported antimicrobial peptides with high binding affinities compared with the native ligand NikZ on the target receptor, the binding energies of five ligands to the target protein being −6.6 (NikZ), −5.4 (AMP_01), −5.5 (AMP_02), −4.7 (AMP_03), and −8.8 (AMP_04) kcal/mol, respectively (Table 1). This indicates that AMP_04 exhibited the highest predicted binding energy of −8.8 kcal/mol, indicating its strongest affinity to the target protein, even better than that of the Phase II clinical stage commercial drug NikZ. This enhanced affinity can be attributed to the molecule’s ability to form favorable interactions with various binding site residues, forming hydrogen bonds with the target protein and resulting in the highest overall binding energy. Thus, these preliminary findings indicate that AMP_04 is a promising candidate with potentially stronger affinity to the target protein compared to other antimicrobial peptides, warranting further validation and exploration.

### 2.2. The Inhibitory Mechanism of AMP_04 on Chitin Synthase

To further elucidate the antimicrobial mechanism of AMP_04, we performed molecular docking studies to examine its interaction with chitin synthase compared to NikZ. Utilizing the cryo-electron microscopy structure of NikZ and chitin synthase, as reported by Yang et al. [35], we found that the uridine moiety of NikZ binds to residues T237, E241, D291, and D382 in the substrate binding pocket of chitin synthase, effectively blocking substrate binding. The hydroxypyridine group of NikZ also partially occupies the catalytic and translocation channels, hindering substrate transfer (Figure 2a).

In contrast, AMP_04 exhibits a distinct binding mode on which its head region, akin to the uridine moiety of NikZ, interacts with residues D291 and D382, occupying the substrate binding pocket. Remarkably, the tail region of AMP_04 penetrates deeper into chitin synthase, occupying the catalytic cavity and translocation channel, and forms hydrogen bonds with amino acid residues, including P454, K537, and R538, enhancing stability. The body region of AMP_04 establishes additional hydrogen bonds with surrounding residues K355, S436, D440, S534, and Y732, further reinforcing the binding configuration (Figure 2b).

In summary, it can be observed that AMP_04 not only resides in the substrate binding pocket of chitin synthase but also extends deeper into the catalytic cavity and translocation channel, where it forms a network of hydrogen bonds with surrounding residues, which may result in AMP_04’s binding to chitin synthase being more robust than that of NikZ. However, NikZ forms hydrogen bond interactions primarily with the substrate binding pocket of chitin synthase.

### 2.3. Saturation Mutagenesis of AMP_04 In Silico

To further enhance the stability and binding affinity of AMP_04 to chitin synthase, we implemented saturation mutagenesis studies on AMP_04 by Discovery Studio incorporating single, double, and triple mutations. The most favorable result, identified after three rounds of mutations, was the triple mutant TP (D10K, G11I, and S14L), which demonstrated a significant improvement in stability and binding affinity.

In the initial round of single mutations, we identified several favorable mutations in AMP_04, including E1Y, S14I, D8I, D10H, A19W, G11P, and A13Q (Figure 3). Despite an increase in mutation energy for most single mutants, hydrogen bond interactions decreased. We selected S14I (SP, mutation energy of −3.47 kcal/mol) as the optimal single mutant, considering all factors.

In the second round, we focused on double mutations. Integrating mutational energy and interaction analysis, we found that the double mutant DP (D10K and S14L, mutation energy of −5.41 kcal/mol) resulted in an impressive enhancement of approximately 50% in mutational energy. Notably, the mutation at amino acid K10 introduced interactions with three residues in the receptor protein (E241, Q245, and V383), with E241 and V383 residing within the substrate-binding pocket of chitin synthase (Appendix A). This mutation allowed K10 to penetrate deeper into the substrate-binding pocket than the D10 residue, effectively occupying the pocket and obstructing substrate entry (Figure 4b).

To further optimize AMP_04, a third round of mutations was conducted based on the K10 and L14 positions. The optimal mutation from this round was TP (D10K, G11I, and S14L, mutation energy of −6.81 kcal/mol), which resulted in nearly double the mutational energy compared to the SP. Additional interactions were observed compared to the double mutants, such as E1 and R538 and A13 and W539 (Appendix A). The W539 residue is located within the catalytic and translocation channel of chitin synthase. The 11I mutation in TP effectively occupies the entire catalytic and translocation channel, thereby entirely blocking substrate entry into this region (Figure 4c). In order to verify the advantages of the mutant, MM_GBSA analysis was performed to calculate the binding free energy of antimicrobial peptides with chitin synthase accurately (Table 2). Through the calculation of MM_GBSA, it can be seen that the triple mutant TP has the best affinity with chitin synthase.

### 2.4. Evaluating Antimicrobial Activity and Toxicity

To elucidate the impact of mutations on the antimicrobial peptide, we conducted a helical wheel analysis. This analysis revealed an increase in the peptide’s hydrophobicity and a corresponding decrease in polarity with escalating mutations (Figure 5). This amplified hydrophobicity may contribute to enhanced membrane permeability, as the triple mutant TP relies on interactions between hydrophobic amino acids and the cell membrane to generate pores.

Our results further demonstrated that the double mutant, DP, significantly amplified its interaction with chitin synthase (Appendix A). Moreover, the triple mutant, TP, augmented this interaction even further relative to DP (Appendix A). Remarkably, TP exhibited nearly triple the hydrogen bond count of the wild type and extended its hydrogen bonding interaction with the V383 amino acid within the substrate-binding pocket compared to DP.

Collectively, these results suggest that after three rounds of saturation mutagenesis, the stability of the antimicrobial peptide was substantially enhanced, and its interaction with chitin synthase was nearly perfectly tailored. The mutated antimicrobial peptide, TP, almost entirely occupies both the substrate-binding pocket and the catalytic and translocation channel of chitin synthase, thus significantly augmenting its inhibitory activity and binding affinity.

In light of these findings, we further investigated the antimicrobial capabilities and toxicity of TP. Utilizing the sAMPpred-GAT model proposed by Yan et al., we predicted the scoring profiles of AMP_04 and its mutants as antimicrobial peptides. Intriguingly, a direct correlation between the number of mutated amino acids and the antimicrobial peptide scores was observed. Most notably, the final triple mutant, TP, achieved the highest antimicrobial peptide score of 0.999 (Table 3) [44]. For toxicity prediction, we employed the advanced approach known as ToxIBTL [45]. Consistently, the results indicated a decline in toxicity with an increasing number of mutations, with TP scoring the highest while exhibiting the lowest toxicity (Table 4).

### 2.5. Frontier Molecular Orbitals (FMOs) Analysis

To assess the biological attributes, chemical reactivity, and molecular stability of antimicrobial peptides, we performed frontier molecular orbitals analysis. The HOMO-LUMO energy difference (ΔE_L-H_) is essential in determining the reactivity and stability of the molecule.

Our findings reveal that the ΔE_L-H_ values for AMP_04, DP, and TP are 5.08605 eV, 5.57248 eV, and 5.18350 eV, respectively (Table 5 and Figure 6). The reactivity descriptors derived from the HOMO and LUMO orbitals, encompassing hardness (η), are detailed in Table 5. The three antimicrobial peptides under investigation demonstrate large ΔE_L-H_ and high hardness, indicative of their excellent stability. Notably, the ΔE_L-H_ in the mutant variants DP and TP is more significant than that in AMP_04. The FMOs analysis suggests that the stability of DP and TP surpasses that of AMP_04.

These findings provide a deeper understanding of the quantum mechanical properties of these antimicrobial peptides, further enhancing our comprehension of their stability and reactivity.

### 2.6. Investigation of the Dynamic Binding State of the Chitin Synthase–Antimicrobial Peptide TP Complex through Molecular Dynamics Simulations

To investigate the dynamic behavior of the interaction between the antimicrobial peptide TP and the chitin synthase complex, molecular dynamics simulations were conducted. The dynamic trajectory of the complex was analyzed to elucidate the interaction mechanism. The analysis of the obtained trajectories revealed that the RMSD values of the complex (Appendix A) reached equilibrium after 80 ns and remained stable without further drift. The RMSD of the antimicrobial peptide TP (Appendix A) stabilized at around 50 ns and exhibited slight oscillations throughout the whole simulation. Furthermore, the structural stability of the complex and compactness of the protein upon binding with the ligand was examined by analyzing RMSF values, the radius of gyration (Rg), and solvent-accessible surface area (SASA). The RMSF analysis of residues using g_rmsf (Appendix A) indicated notable fluctuations in the first 100 residues. The radius of gyration (Appendix A) illustrated the increasing compactness of the complex over time, reaching its tightest state at around 70 ns. Furthermore, the investigation of the solvent-accessible surface area (Appendix A) confirmed this finding, as the solvent-exposed surface area of the complex decreased progressively with time, reaching its minimum at approximately 70 ns. Based on the computational analysis of RMSD, RMSF, radius of gyration, and solvent-accessible surface area, it can be inferred that the Chitin synthase–TP complex as a whole is relatively stable and exhibits tight binding.

### 2.7. The Transmembrane Mechanism of Antimicrobial Peptide TP

While the cell wall is the initial porous barrier that permits the movement of substances between the cell and its environment, it allows antimicrobial peptides to enter the periplasmic space or area just outside the cell membrane in certain organisms [46]. However, the cell membrane acts as the main barrier, preventing these peptides from entering the cell’s interior. Consequently, antimicrobial peptides must overcome the cell membrane’s defenses to reach intracellular targets and fulfill their antimicrobial function. Briolay et al. found that phosphatidic acid (PA), phosphatidylcholine (PC), phosphatidylserine (PS), and PtdIns are components of the *Phytophthora* plasma membrane [47,48]. Consequently, a mixture of PA and PC phospholipids was chosen to construct the cell membrane.

All-atom molecular dynamics simulations were performed to elucidate the transmembrane mechanism of antimicrobial peptides at the molecular level. Our findings suggest the mechanism by which antimicrobial peptides penetrate the cell membrane is akin to a barrel-stave mechanism (Figure 7).

According to this study, when the peptide reaches the necessary condition for membrane insertion, it exerts its antimicrobial effect by traversing the membrane. The initial state at 0 ps shows the peptide in a perpendicular, linear structure, representing the initial conformation of the antimicrobial peptide. At 100 ps, the peptide closely adheres to the upper layer of the membrane and alters its conformation, lying flat on the membrane with its hydrophobic amino acids (4V, 5G, 6L) approaching the membrane. By 300 ps, the peptide begins to form a pore on the cell membrane, inserting its hydrophobic amino acids into the cell membrane and creating a conduit for peptide entry. At 500 ps, most of the peptide has entered the cell membrane. By 800 ps, the peptide approaches the lower layer of the membrane, the hydrophobic amino acids are reinserted into the membrane surface, and the peptide transverses by forming a pore. At 1200 ps, the peptide penetrates the lower layer of the cell membrane, with hydrophobic amino acids crossing the cell membrane, presenting a linear transmembrane pattern (Figure 8a).

We examined the question of which segment of the N-terminus and C-terminus of the antimicrobial peptide, TP, approaches the cell membrane first. From a kinetic perspective, the conformation of the N-terminal close to the cell membrane takes slightly less time to cross the cell membrane than the conformation of the C-terminal close to the cell membrane. Therefore, the conformation of the N-terminus close to the cell membrane seems to be more advantageous (Appendix A). Moreover, whether the N-terminus is close to the cell membrane or the C-terminus is close to the cell membrane, the hydrophobic amino acids L and G seem to play an important role in the membrane penetration process. At 0 ps, the C-terminal conformation close to the cell membrane is placed 2 nm above the cell membrane. At 300 ps, the antimicrobial peptide approaches the cell membrane in a suitable conformation, as does the hydrophobic amino acid (17L, 18G). At 500 ps, the antimicrobial peptide begins to pass through the cell membrane. At 600 ps, the antimicrobial peptide enters the cell membrane. At 1000 ps, the antimicrobial peptide approaches the second layer of phospholipid molecules. At 1400 ps, the antimicrobial peptide penetrates the second layer of phospholipid molecules (Figure 8b).

During the process of penetration, three specific polar charged amino acids—1E, 8D, and 10K—come into contact with hydrophobic fatty acid chains. This crucial interaction has prompted us to closely examine the transmembrane dynamics of these amino acids. The intricacies of these interactions are illustrated in Figure 9. Our findings clearly indicate that upon first encountering the cell membrane, the polar nature of these amino acids leads to a strong affinity for water molecules and the hydrophilic portions of the phospholipid heads. As the transmembrane permeation progresses, these amino acids persistently associate with the hydrophilic head regions while consistently avoiding the hydrophobic tail regions due to their inherent polarity.

## 3. Discussion

Through virtual screening, we have identified the optimal lead molecule AMP_04 based on its binding affinity to chitin synthase. Subsequently, we conducted molecular docking and molecular dynamics simulations to investigate the mechanism of binding between the antimicrobial peptide and chitin synthase compared with the commercially available drug NikZ. Interestingly, in silico analysis indicates that AMP_04 almost entirely occupies the substrate-binding pocket and catalytic transport channel of chitin synthase. To further optimize the affinity between AMP_04 and chitin synthase, saturation mutagenesis exploration was performed, resulting in the mutant variant TP (D10K, G11I, S14L) [49]. Finally, to elucidate the transmembrane mechanism of the antimicrobial peptide TP, we conducted dynamic simulations to demonstrate how TP achieves antimicrobial action by translocating through the cell membrane using an instantaneous membrane rupture. The hydrophobic amino acids of antimicrobial peptides approach the cell membrane surface. Similar to the barrel-stave mechanism, they form pores that enable the peptides to penetrate the cell membrane. They might penetrate the biomembrane and then bind to chitin synthase, which is crucial for cell survival. The mode of intracellular binding involves inhibiting the growth and reproduction of *P. sojae* by interacting with and suppressing the activity of chitin synthase [50]. This mechanism utilizes hydrophobic amino acids that interact with the surface of the cell membrane, forming pores for antimicrobial effects [51]. The optimized antimicrobial peptide TP, targeting chitin synthase, might not only address the diseases in soybeans caused by *P. sojae* but also overcome concerns regarding the safety of genetically modified crops and the problem of drug resistance in small-molecule drugs.

In this study, we have designed antimicrobial peptides based on empirical validations and have interpreted their inhibitory mechanisms on chitin synthase and membrane penetration at the molecular level. Current strategies to control soybean *Phytophthora* rely heavily on genetically modified crops and pesticides, which pose substantial environmental and health hazards. Genetically modified soybeans exacerbate global warming and promote water and land acidification while potentially escalating cancer risks. Pesticides, with their persistent and toxic nature, can detrimentally impact ecosystems and human health. In contrast, antimicrobial peptides possess distinct advantages, including diverse killing mechanisms, broad-spectrum efficacy, and reduced host cell toxicity. The development of environmentally benign and safe antimicrobial peptides is, therefore, a critical step towards sustainable soybean *Phytophthora* management. Our research findings offer invaluable insights into the prospective application of these ligands as antimycotic agents.

One potentially effective way to combat these pathogens is to target the biosynthesis of chitin. A promising path toward the development of environmentally friendly antimicrobial compounds is the inhibition of chitin synthase, an enzyme essential to the various pathogens that is absent in plants and mammals. In the design of chitin synthase inhibitors, the configuration of the active site is pivotal. We employed AutoDock Vina docking, given the involvement of the catalytic pocket and the transport catalytic channel in the chitin synthesis process [52]. Perhaps some of the latest methods that other researchers have recently explored to improve on docking tools can be used, such as force field improvements, conformation search algorithm improvements, and scoring function improvements [53,54,55,56,57,58,59,60,61,62,63]. Within the catalytic pocket, residue D496 functions as the catalytic residue, facilitating nucleophilic addition via protonation, while residue E495 assists in substrate recognition and alignment [64]. Inhibiting these key amino acids in the catalytic pocket effectively impedes catalysis. Within the transport channel, VLPGA_452–456_ operates as a gatekeeper, regulating the channel’s opening and closing for product egress. Inhibiting key amino acids in the transport channel effectively halts chitin elongation. A critical amino acid, D291, interacts with both chitin synthase and NikZ and is integral for substrate binding [65]. These key residues provide a theoretical foundation for the future development of chitin synthase inhibitors, indispensable for designing novel anti-phytophthora compounds [66].

Regarding the stability of antimicrobial peptides, it is acknowledged that many exhibit poor stability. Our approach seeks to regulate the stability of antimicrobial peptides by manipulating the HOMO-LUMO energy gap via electronic effects [67]. This method can serve as a valuable reference for designing stable molecules [68,69,70]. Our observations indicate that the HOMO and LUMO orbitals of antimicrobial peptides predominantly reside on the peptide bonds of the amino acids. The HOMO, representing the ability to donate electrons, typically occupies positions near peptide bonds with electron-donating groups. Conversely, the LUMO, signifying the ability to accept electrons, usually resides near peptide bonds with electron-withdrawing groups. By introducing electron-donating groups in the HOMO orbital or electron-withdrawing groups in the LUMO orbital, it is possible to augment the energy gap, which may enhance the stability of antimicrobial peptides [71,72,73,74].

Antimicrobial Peptides have shown considerable promise as sustainable and eco-friendly alternatives to antibiotics in agriculture. Their multifunctional capabilities extend beyond direct antimicrobial action to include antioxidant, immunomodulatory, and anticancer activities, thereby offering potential improvements in livestock health and production performance, as well as plant protection [75]. Moreover, AMPs are emerging as innovative solutions for water purification in agricultural ecosystems and as benign agents for marine life [76,77]. The commercial success of several AMPs as food preservatives also highlights their role in enhancing the safety and shelf life of agri-products [78]. Future research is encouraged to assess further the environmental impacts and practical applications of AMPs in various agricultural contexts.

Our approach to designing anti-phytophthora peptide compounds could hold significant implications for the discovery and optimization of future antimicrobial peptides. The use of computational methods for effective drug design is an irreversible trend as it facilitates a comprehensive understanding of the structure and binding mode between target proteins and ligands, which is crucial for optimizing antimicrobial peptides [79,80,81]. Our design theory may guide others by directing the saturation mutagenesis of antimicrobial peptides, focusing on both electronic effects and spatial structures. Maximizing interactions between antimicrobial peptides and key amino acids in target proteins through electronic effects, and occupying critical structural domains through spatial structures, are essential [82,83,84]. Furthermore, membrane penetration is vital for antimicrobial peptides, and understanding the membrane translocation mechanism through stretching dynamics or umbrella sampling can provide valuable insights, as our understanding of peptide translocation mechanisms remains limited [85,86,87].

## 4. Materials and Methods

### 4.1. Preparation of Protein Receptor and Ligands

The crystal structure of the protein chitin synthase (PDB ID: 7WJO) was obtained from the RCSB database (https://www.rcsbpdb.org (accessed on 6 June 2023)). Further, the target protein was optimized using Discovery Studio v4.5 (BIOVIA Corp, San Diego, CA, USA). In the preparation and optimization process, the associated co-crystallized ligand (NikZ) and all the water molecules were removed from the protein molecule [88,89,90]. The chains A and B (chitin synthase) were selected for further analysis, and the missing atoms of the incomplete amino acid side-chain or backbone and the hydrogen atoms were added to the 3D structures of protein by using Discovery Studio v4.5.

Moreover, the preprocessing of four antimicrobial peptides was performed using Discovery Studio v4.5 software. The 3D crystal structures of four antimicrobial peptides were predicted using ColabFold v1.5.2: AlphaFold2 using MMseqs2 (https://colab.research.google.com/github/sokrypton/ColabFold/blob/main/AlphaFold2.ipynb (accessed on 6 June 2023)) [91]. AlphaFold2 (AF2) is an innovative protein structure prediction method that combines deep learning and physical modeling [92]. It utilizes the primary amino acid sequence to predict the coordinates of all heavy atoms in a given protein. By incorporating protein sequence contextual information and leveraging evolutionary and physical constraints, AF2 achieves exceptional accuracy in predicting protein structures. Its workflow involves unsupervised learning from large-scale protein structure databases, collaborative learning for multiple sequence alignment, and pairwise amino acid comparison. AlphaFold2’s ability to accurately generate three-dimensional protein structure predictions, even in complex scenarios and without complete physical environment information, sets it apart as a groundbreaking approach in the field. The quality of the model is evaluated through the parameter of predicted local distance difference test (pLDDT), which measures the confidence of the model at each position, on a scale of 0 to 100. By comparing pLDDT, the 3D crystal structures of the four antimicrobial peptides are obtained (Appendix A).

### 4.2. Virtual Screening

The virtual screening of antimicrobial peptides with chitin synthase was performed using AutoDock Vina-1.2.2 [93]. PDBQT files prepared by AutoDock Tool 1.5.6 were used to dock ligands to the receptor’s binding site. The grid box was generated around key amino acid residues, including L412, Y433, V452, P454, and W539. These amino acids have been reported in the literature to be significantly correlated with chitin synthase catalytic activity and were included in the docking box setup [35]. The box size was defined with center_x = 20.3 Å, center_y = 13.2 Å, center_z = 15.7 Å, size_x = 76.0 Å, size_y = 62.0 Å, and size_z = 80.0 Å. For a global search, the exhaustiveness was set to 100, which was sufficient to achieve reliable results. AutoDock Vina utilizes a novel scoring function that incorporates contributions from various energy terms, including van der Waals forces, hydrogen bonding interactions, and electrostatic interactions. The docking results were analyzed for interaction using “Analyze Protein Interface” in Discovery Studio v4.5.

### 4.3. Calculation of Mutation Energy (Binding) for Saturation Mutagenesis

The saturation mutagenesis analysis and computation of mutation energy were performed by using the “Design Protein” module of Discovery Studio v4.5 [94,95,96,97,98]. The energy effect of each mutation on the binding affinity (mutation energy, ΔΔGmut) is calculated in the CHARMm Polar H force field and pH-dependent mode. The pH value, ionic strength, solvent dielectric constant, and energy cutoff were set at 7.4, 0.1, 80, and 0.5, respectively. The mutation energy represents the stability of the chitin synthase–antimicrobial peptide complexes. The lower mutation energy indicates the more stable structure of the chitin synthase–antimicrobial peptide complex. In the context of protein saturation mutagenesis research, pH-dependent electrostatic interactions are considered during simulations, while temperature-related calculations are excluded. The system is accurately described using the CHARMm force field, combining an effective dielectric constant of 10 and a solvent dielectric constant of 80. Initial minimization techniques are applied to enhance efficiency and accuracy, limiting the maximum number of preserved structures to 20 and the maximum number of mutations to 1,000,000. The heating map was generated using Excel 2021, while the interaction map was created using PyMoL-2.5.5.

### 4.4. Helical Analysis of Antimicrobial Peptides

The properties of antimicrobial peptides were analyzed using the HeliQuest online server (https://heliquest.ipmc.cnrs.fr/ (accessed on 28 September 2023)), which is capable of calculating and analyzing the physicochemical properties of amino acid sequences. This tool can be utilized for database screening and the design of amino acid sequences with similar functionality but distinct sequences [99]. We utilized the HeliQuest server to calculate the hydrophobicity, polarity, and charge of peptide sequences. Substitutions of amino acids within the peptide will result in alterations to the peptide’s properties, thereby influencing antimicrobial activity, toxicity, and membrane permeability. By analyzing the sequence–structure relationship, it is possible to identify the amino acids that have a significant impact on antimicrobial activity, toxicity, and membrane permeability. The HeliQuest server is capable of determining the properties of an amino acid sequence. Initially, the α-helix sequence submitted by the user is subjected to sliding window analysis, with a window size of 18 amino acids. By calculating the average hydrophobicity, hydrophobic moment, net charge, and the content of different types of polar amino acids within each window sequence, a detailed profile of the sequence’s physical and chemical attributes is obtained. Furthermore, the HeliQuest service can also predict the secondary structure of the amino acid sequence. It integrates three methods, namely TMHMM, PSIPRED, and linear discriminant analysis, to predict the secondary structure of the sequence. Additionally, by setting filtering criteria based on the calculated attributes of the reference sequence, the database is analyzed to identify potential α-helix sequences with similar attribute features.

### 4.5. 2D Interaction Graph Analysis

To visualize the interactions between ligands and binding site residues, LigPlot+ (version v.2.2) was employed [100]. You can access and download it through https://www.ebi.ac.uk/thornton-srv/software/LigPlus/ (accessed on 28 September 2023). By depicting diagrams illustrating hydrogen bonding, hydrophobic interactions, and non-hydrogen bonding interactions, clear and visually appealing 2D graphics can be obtained to identify the residues that contribute to the stability of the complex. LigPlot+ can automatically load structural data from multiple PDB files. It identifies equivalent residues in different structures through sequence and structure alignment and compares multiple protein-ligand structures in a 2D graphical representation. The first structure serves as a reference, and the ligand molecules in the subsequent structures are rotated and flipped to align equivalent atoms while preserving the corresponding residue positions in 2D. LigPlot+ enables the visualization of ligand binding mode differences and similarities between various structures. LigPlot+ is a valuable tool for researchers to quickly assess changes in binding sites, providing essential insights for drug design and mechanism studies.

### 4.6. MM_GBSA Analysis

The MM_GBSA binding free energy calculations were carried out using Maestro v9.0 (Schrodinger, LLC, New York, NY, USA). The more negative the MM_GBSA value, the better the affinity between the antimicrobial peptide and chitin synthase. The complexes were refined with Prime under the OPLS3 force field, adopting the Variable Dielectric Surface Generalized Born (VSGB) continuum solvation model [97]. The energies obtained for the complexes were automatically calculated based on the energy terms and the equation systems reported in the following:$$\Delta G_{binding}=G_{complex}-(G_{receptor}+G_{ligand})$$
$$\Delta G_{binding}=\Delta E_{MM}+\Delta G_{GB}+\Delta G_{SA}$$
$$\Delta E_{MM}=\Delta E_{internal}+\Delta E_{electrostatic}+\Delta E_{vdw}$$

The overall binding free energy, $\Delta G_{binding}$, is calculated by considering the energies of the complex, the receptor, and the ligand separately. This total binding energy comprises the molecular mechanics (MM) energies, including bond, angle, and torsional strain ($\Delta E_{internal}$), electrostatic ($\Delta E_{electrostatic}$), and van der Waals ($\Delta E_{vdw}$) interactions, as well as contributions from the solvation energies, which account for both polar ($\Delta G_{GB}$) and nonpolar ($\Delta G_{SA}$) effects [101,102,103].

### 4.7. Prediction of the Activity and Toxicity of Antimicrobial Peptides

The sAMPpred-GAT graph neural network model can be accessed at http://bliulab.net/sAMPpred-GAT (accessed on 9 August 2023) [104]. The model used in this study is a graph attention network-based AMP prediction model. It constructs a graph structure using peptide sequences and structural information and then utilizes the GAT network to learn node features for AMP classification prediction. Experimental results demonstrate that this model outperforms other methods in terms of predictive performance across multiple datasets.

The toxicological prediction model ToxIBTL can be accessed at https://server.wei-group.net/ToxIBTL/Server.hTPl (accessed on 9 August 2023) [45]. ToxIBTL is a cutting-edge deep learning framework that seamlessly integrates the principles of the information bottleneck and transfer learning, enabling precise prognostication of peptides and proteins’ toxicity. Through the fusion of evolutionary insights, physicochemical characteristics, and feature representation learning, ToxIBTL not only achieves unparalleled accuracy in peptide datasets but also demonstrates commendable competitiveness in protein datasets.

### 4.8. Quantum Chemical Analysis

The chemical quantum study investigated the chemical stability of the best three antimicrobial peptides, including the analysis of frontier molecular orbitals (FMO). The theoretical chemical quantum calculations for the molecular structural analysis of selected antimicrobial peptides were conducted using GAUSSIAN 09 [105]. The DFT analysis was performed to examine the electronic structure properties of the top-hit ligand molecules. Additionally, the visualization of frontier molecular orbitals was performed using Multiwfn version 3.8 and VMD version 1.9.3 software [106,107]. For quantum chemical calculations of antimicrobial peptides, their chemical structures were optimized (the lowest energy conformation) using gradient-corrected (density functional theory) DFT with the three-parameter hybrid functional (B3) for the exchange part, and the Lee-Yang-Parr (LYP) correlation function was utilized to compute the molecular structure, vibrational frequencies, and energies of the optimized structures.

Moreover, to further explain the dispersion interactions that the B3LYP function is unable to describe, B3LYP-D3 was employed. Meanwhile, the basis set 6−31+G(d,p) was augmented by polarization functions on heavy atoms, polarization functions on hydrogen atoms, and diffuse functions for both hydrogen and heavy atoms. The optimized geometries have been used to calculate the HOMO and LUMO energy parameters in this study [108,109]. The hardness (η) is computed based on Koopman’s theory [110]. The formula is as follows:$$\eta =\frac{1}{2}(E_{LUMO}-E_{HOMO})$$

### 4.9. Molecular Dynamics Simulations

Based on the optimal structures obtained by docking experiments, the antimicrobial peptides-protein molecular dynamics simulations were performed with the AMBER99 corrected force field (AMBER99SB-ILDN) using the GROMACS 2020.01 software package in the Ubuntu 20.04 environment [111]. This simulation aimed to explore the atomic behavior, protein folding, and conformational changes within the dynamic environment of the protein. The topological structure of the complex was generated using the pdb2gmx command, with the AMBER99SB-ILDN force field chosen [112]. Energy minimization was executed using the steepest descent method, with a maximum force cutoff of 1000 kJ/mol. To create a solvation environment, the water model of choice is SPC216, and the complex is positioned within a cubic box system with dimensions extending 1.0 nm from the edges of the system [113]. Sodium ions were added to neutralize the system’s charge, and periodic boundary conditions were applied in all directions. Electrostatic interactions were computed using the Particle Mesh Ewald (PME) method, with a cutoff distance of 1 nm for short-range electrostatic interactions. Van der Waals interactions were calculated using a cutoff truncation at 1 nm. The LINCS algorithm was employed for constraints. A temperature equilibration was carried out for a continuous period of 100 ps at a temperature of 300 K and a coupling constant of 0.1 ps, followed by a pressure equilibration at a constant pressure of 1 bar, with a coupling constant of 2 ps. The molecular dynamics simulation was then run for 100 ns to facilitate energy and trajectory analysis. Trajectory analysis of the complex was performed using g_rmsd, g_rmsf, g_gyrate, and gmx_sasa. The obtained results were further analyzed using the Xmgrace-5.1.25 tool.

### 4.10. Dynamic Simulation of Membrane Penetration Mechanism

We constructed a peptide-membrane system using CHARMM-GUI (https://charmm-gui.org/ (accessed on 23 January 2024)) [114]. In the Membrane Builder module, we submitted a peptide pdb file and positioned the peptide 2 nm above the first layer of the membrane [115]. The upper membrane layer consisted of 38 POPA molecules and 38 POPC molecules. The lower layer contained 39 POPA molecules and 39 POPC molecules. We added 106 Na^+^ ions and 28 Cl^−^ ions using a Monte Carlo-based method. The system was equilibrated at a temperature of 303.15 K. Force field was selected as AMBER19SB [116]. After successful system construction, energy minimization was performed, followed by six rounds of equilibration steps using the GROMACS software package. Once the system reached equilibrium, we conducted a 2000 ps membrane crossing molecular dynamics simulation. We employed the leap-frog algorithm with a time step of 0.002 fs, the particle-mesh Ewald method for calculating electrostatic interactions, and a cutoff for van der Waals interactions. The LINCS algorithm was used to constrain bond lengths, and an umbrella potential was employed to pull the peptide center of mass downwards along the *z*-axis at a rate of 0.005 nm/ps, applying a force constant of 1500 kJ/(mol·nm^2^). After completing the molecular dynamics simulation, we used the trjconv command to extract structural files from the trajectory. The obtained structures were visualized and analyzed using PyMoL-2.5.5.

## 5. Conclusions

In conclusion, our study has successfully utilized computational design to create TP, a novel antimicrobial peptide that significantly inhibits chitin synthase—a target enzyme exclusive to pathogens and not found in plants or mammals, thus constituting an eco-friendly option. This peptide demonstrates robust engagement with the chitin synthase’s substrate binding site and catalytic channel, suggesting effective enzyme interference. From our verified peptide library, AMP_04 was selected and subjected to saturation mutagenesis, enhancing its antimicrobial effectiveness, reducing toxicity, and bolstering stability. The optimized TP mutants exhibit high activity without being harmful to human cells. Furthermore, TP’s translocation into membranes likely operates via a barrel-stave model, primarily driven by its hydrophobic amino acids. Our findings offer a detailed look at the mechanism by which TP impedes chitin synthase, and underscore its potential as a sustainable tool against *P. sojae*, promising advances toward greener agricultural methods.

## Figures and Tables

**Figure 1 ijms-25-03512-f001:**
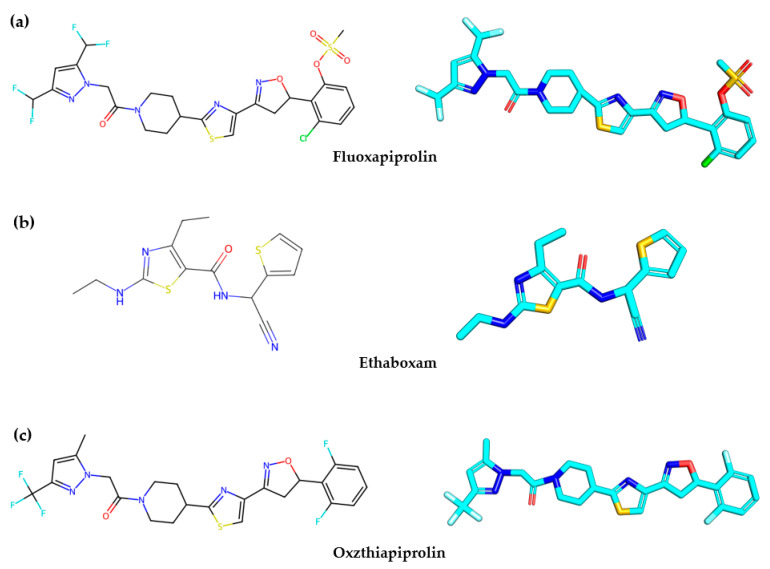
Drugs that inhibit soybean phytophthora blight. (**a**) Fluoxapiprolin, (**b**) Ethaboxam, (**c**) Oxzthiapiprolin.

**Figure 2 ijms-25-03512-f002:**
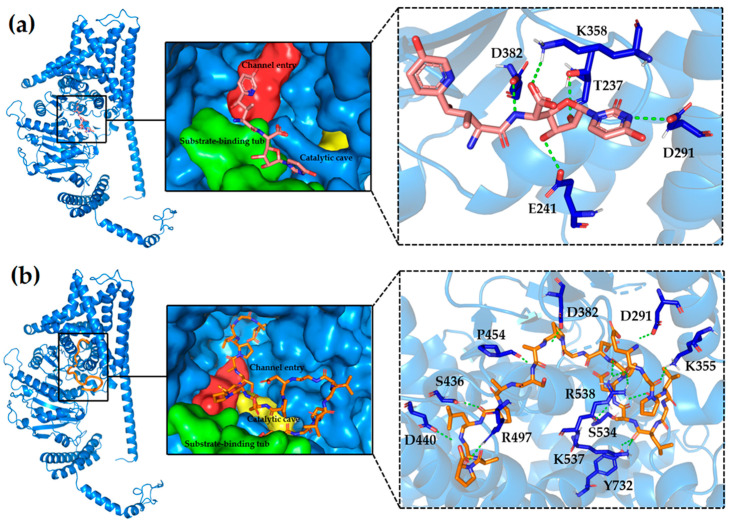
Mechanism of interaction between different ligands and chitin synthase. (**a**) Mechanism of inhibition of chitin synthase acted on by commercial drug NikZ, (**b**) mechanism of inhibition of chitin synthase by AMP_04. In the surface representations, the green area represents the substrate-binding tub, the yellow area represents the catalytic cave, and the red area represents the channel entry.

**Figure 3 ijms-25-03512-f003:**
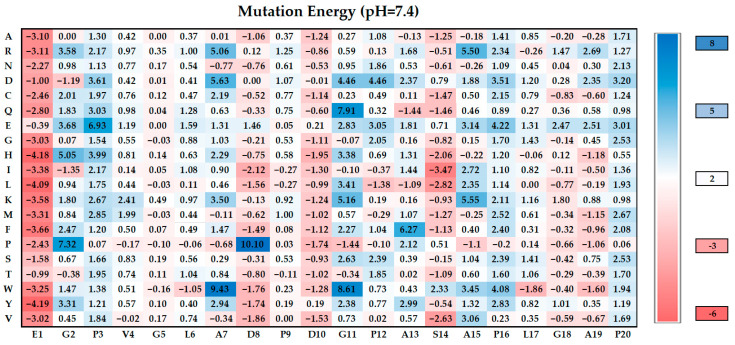
Mutation energy mapping of the AMP_04 antimicrobial peptide single mutants. Each letter on the left vertical axis represents a type of amino acid after mutation.

**Figure 4 ijms-25-03512-f004:**
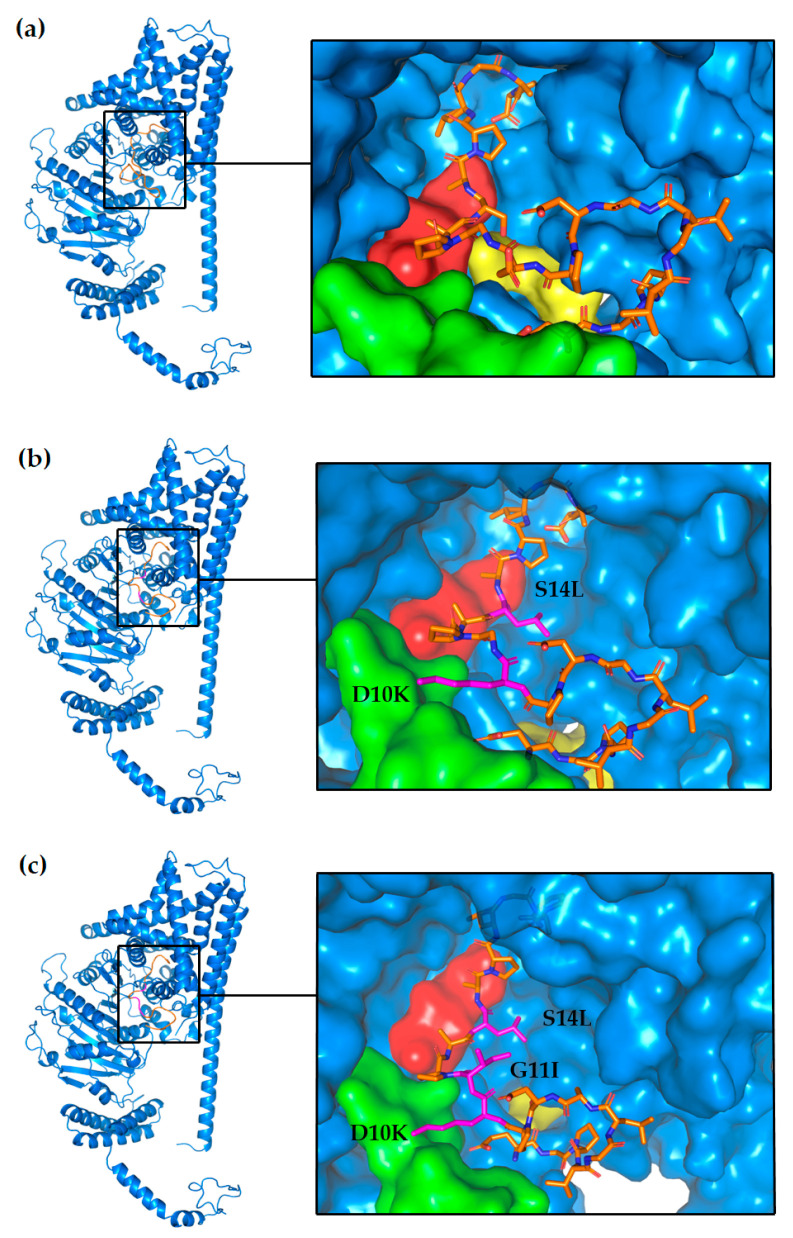
The spatial interaction diagram of antimicrobial peptide AMP_04 and its mutants with chitin synthase. (**a**) Interactions between wild-type AMP_04 and chitin synthase; (**b**) interactions between DP double mutant and chitin synthase; (**c**) interactions between TP triple mutant and chitin synthase. In the surface representations, the green area represents the substrate-binding tub, the yellow area represents the catalytic cave, and the red area represents the channel entry. In the structural representation of antimicrobial peptides, orange represents unmutated amino acids, while the pink color represents the mutant amino acid.

**Figure 5 ijms-25-03512-f005:**
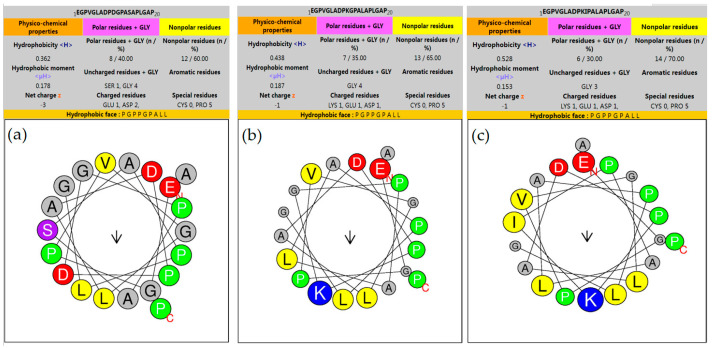
Helical wheel analysis of the antimicrobial peptide and its mutants. (**a**) Antimicrobial Peptide (AMP_04); (**b**) double mutant (DP); (**c**) triple mutant (TP). Red, blue, and gray colors represent anionic, cationic, and hydrophilic amino acids, respectively. Green signifies hydrophobic amino acids. Arrows denote the helical hydrophobic moment. The red letter “N” indicates the peptide’s head, while the red letter “C” marks its tail.

**Figure 6 ijms-25-03512-f006:**
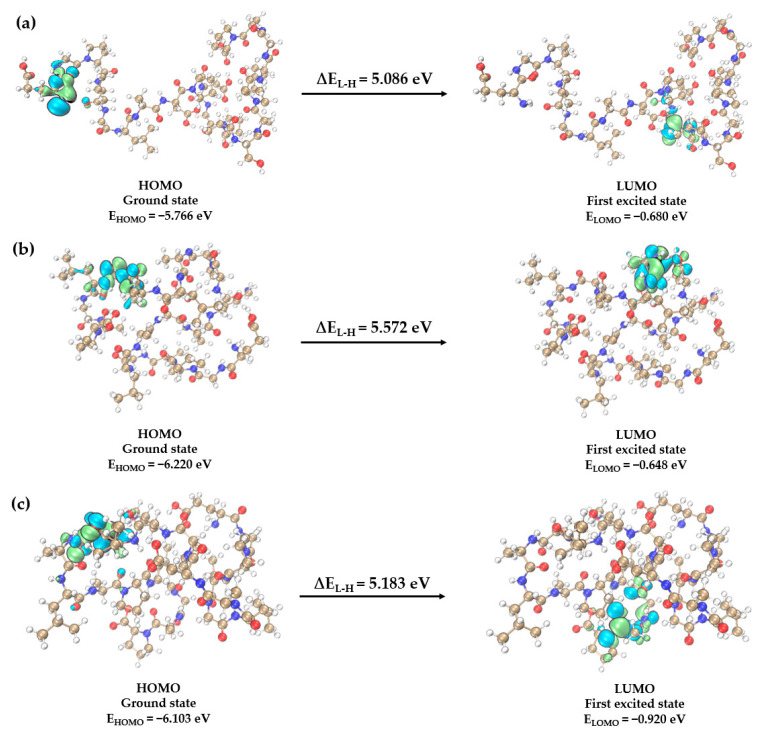
The FMOs, including HOMO and LUMO, for (**a**) AMP_04, (**b**) DP, (**c**) TP, as calculated at B3LYP−D3/6−31+G (d,p) level of DFT. The carbon (C), hydrogen (H), oxygen (O), and nitrogen (N) atoms are colored in tan, white, red, and blue, respectively. The blue and green parts represent the cloud density of frontier orbitals.

**Figure 7 ijms-25-03512-f007:**
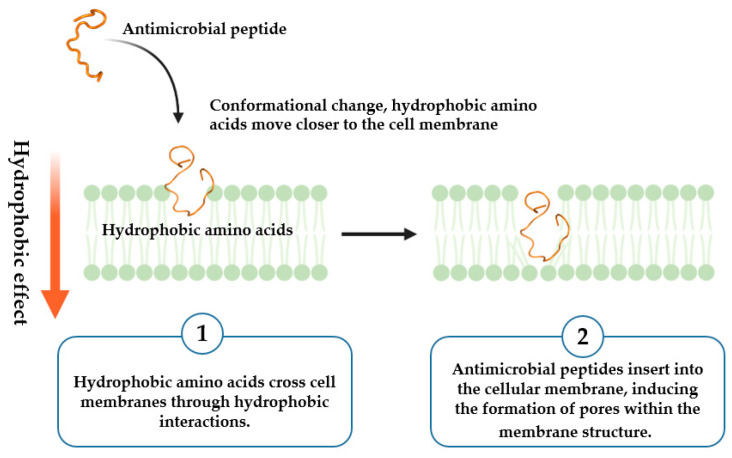
The transmembrane mechanism of antimicrobial peptide TP.

**Figure 8 ijms-25-03512-f008:**
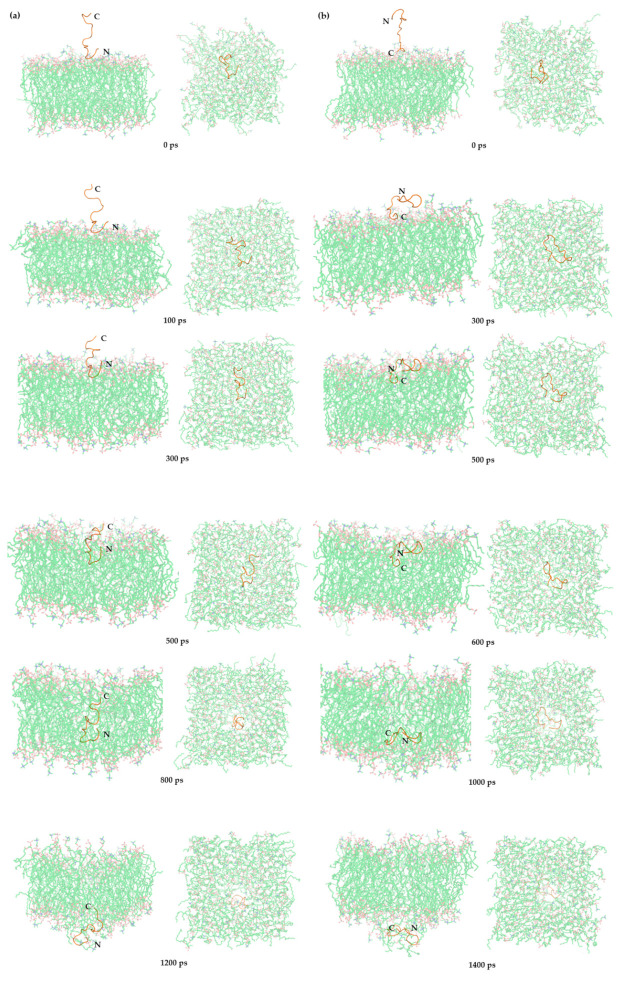
Translocation process diagram of the antimicrobial peptide (TP), with the (**a**) N-terminus and the (**b**) C-terminus approaching the surface of the cell membrane.

**Figure 9 ijms-25-03512-f009:**
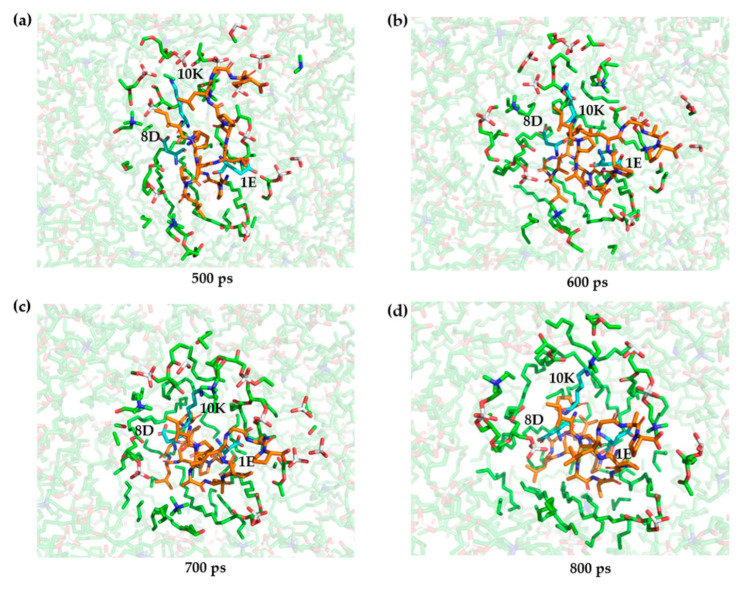
Schematic illustration of the antimicrobial peptide (TP) translocation process, where the polar amino acids 1E, 8D, and 10K undergo changes at (**a**) 500 ps; (**b**) 600 ps; (**c**) 700 ps; (**d**) 800 ps. Blue represents polar amino acids, green signifies the hydrophilic head groups of phospholipid molecules, and orange denotes the antimicrobial peptide molecules.

**Table 1 ijms-25-03512-t001:** The predicted binding energy (kcal/mol) of different ligands with chitin synthase.

Ligand	Amino Acid Sequence	Predicted Binding Energy (kcal/mol)	Reference
NikZ	-	−6.6	[35]
AMP_01	NH_2_-KWKVFKKIEKMGRNIRNGIVKAGPAIAVLGEAKAL-COOH	−5.4	[41]
AMP_02	NH_2_-GIFSKLAGKKLKNLLISGL-COOH	−5.5	[39]
AMP_03	NH_2_-MASRAARLAARLARLALRAL-COOH	−4.7	[42]
AMP_04	NH_2_-EGPVGLADPDGPASAPLGAP-COOH	−8.8	[43]

**Table 2 ijms-25-03512-t002:** Antimicrobial peptides’ binding ability to chitin synthase calculated by MM_GBSA.

Antimicrobial Peptides	MM_GBSA Binding Energies (kcal/mol)
AMP_04	−25.44
DP	−55.11
TP	−90.38

**Table 3 ijms-25-03512-t003:** Predicting the antimicrobial activity of AMP_04 and its mutants using sAMPpred-GAT.

Name	Class	Probability
AMP_04	AMP	0.998
DP	AMP	0.999
TP	AMP	0.999

**Table 4 ijms-25-03512-t004:** Evaluating the toxicity of antimicrobial peptide AMP_04 and its mutants using ToxIBTL.

Name	Results	Score
AMP_04	Non-toxic	$6.631\times 10^{-21}$
DP	Non-toxic	$1.954\times 10^{-12}$
TP	Non-toxic	$1.383\times 10^{-4}$

**Table 5 ijms-25-03512-t005:** HOMO-LUMO energy gap and other physical relationships for antimicrobial peptides.

Parameter	AMP_04	DP	TP
HOMO (eV)	−5.766	−6.220	−6.103
LUMO (eV)	−0.680	−0.648	−0.920
ΔE_L-H_ (eV)	5.086	5.572	5.183
η	2.543	2.786	2.591

## Data Availability

Data is contained within the article and Appendix A. Our code and data are available at https://github.com/rain242/AMP_article_1 (accessed on 17 February 2024).

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
