# Peer review of "Rationally Designed Novel Antimicrobial Peptides Targeting Chitin Synthase for Combating Soybean Phytophthora Blight"

_ijms, 2024, doi:10.3390/ijms25063512_

Round 1

Reviewer 1 Report

Comments and Suggestions for Authors

I have read the manuscript ” Rationally Designed Novel Antimicrobial Peptides Targeting Chitin Synthase for Combatting Soybean Phytophthora Blight”, and I consider the article submitted is interesting.

However, few corrections are required. Provided below are my comments and suggestions.

Line 15 ” evaluate their antimicrobial effects against chitin synthase” I consider that the statement needs to be re-evaluated and requires rephrasing. I consider that the antimicrobial effect is not against chitin synthase.

Line 39 metric tons (do not use capital letters).

Line 68-69 Please give more details about mechanisms of action for killing pathogens. Please, avoid lumping the references [14-17]. Every cited reference should be discussed in the paper.

Line 76 - it would be interesting to have the chemical structures of the mentioned substances in a scheme.

Line 110 Please give more explanation, avoid lumping the references

Line 111-132  I think that this part fits better in the Discussion part of the article. In the Introduction, the authors should briefly describe the research protocol.

Line 454 The expression ”These amino acids have been reported in the literature” requires references from literature.

 Minor observations:

All Latin terms in the manuscript including the references list should be presented in italics (ex. in silico).

The text should also be revised from the point of view of the formulation of the sentences (ex. lines 108-109 )

Comments on the Quality of English Language

The quality of the English language is fine. There are some statements that can use more clarity (noted in my suggestions). 

Reviewer 2 Report

Comments and Suggestions for Authors

Yue Ran and colleagues designed and analysed in silico novel peptide inhibitors of chitin synthase from Phytophthora sojae. P. sojae is a soil borne oomycete pathogen that causes stem and root rot of soybean, often leading to large economic losses. Therefore, developing an effective anti-oomycete agent that would not pose a threat to plants and human health is very important. The manuscript presents comprehensive and convincing in silico results. I have only minor comments:

 Line 56-57. The authors wrote that “genetically modified soybeans could result in higher rates of cancer and tumor occurrence” citing the paper by Séralini et al. (2012) doi:10.1016/j.fct.2012.08.005. The article, in which the effects of genetically modified maize, not soybean, were studied, provoked serious discussion; the results were questioned. The paper was finally retracted from FCT but then resubmitted in 2014 in Environmental Sciences Europe (doi: 10.1186/s12302-014-0014-5).

Line 238, please check the refence: Liu et al. or Yan et al.?

Figure 4, please explain in the legend what type of amino acid each color represents.

Line 370,  should be“…is absent…”

Line 463, should be“Calculation of…”

Round 2

Reviewer 1 Report

Comments and Suggestions for Authors

I believe that the authors have taken into account all my comments and they made all the changes requested so that the article can be published.